# Astaxanthin Supplementation as a Potential Strategy for Enhancing Mitochondrial Adaptations in the Endurance Athlete: An Invited Review

**DOI:** 10.3390/nu16111750

**Published:** 2024-06-03

**Authors:** Hunter Waldman

**Affiliations:** Department of Kinesiology, University of North Alabama, Florence, AL 35630, USA; hswaldman@una.edu

**Keywords:** astaxanthin, sports nutrition, endurance performance, antioxidants, oxidative stress

## Abstract

Astaxanthin, a potent antioxidant found in marine organisms such as microalgae and krill, may offer ergogenic benefits to endurance athletes. Originally used in fish feed, astaxanthin has shown a greater ability to mitigate various reactive oxygen species and maintain the structural integrity of mitochondria compared to other exogenous antioxidants. More recent work has shown that astaxanthin may improve: (1) cycling time trial performance, (2) cardiorespiratory measures such as submaximal heart rate during running or cycling, (3) recovery from delayed-onset muscle soreness, and (4) endogenous antioxidant capacity such as whole blood glutathione within trained populations. In this review, the history of astaxanthin and its chemical structure are first outlined before briefly describing the various adaptations (e.g., mitochondrial biogenesis, enhanced endogenous antioxidant capacity, etc.) which take place specifically at the mitochondrial level as a result of chronic endurance training. The review then concludes with the potential additive effects that astaxanthin may offer in conjunction with endurance training for the endurance athlete and offers some suggested practical recommendations for athletes and coaches interested in supplementing with astaxanthin.

## 1. Introduction

### 1.1. Sports Nutrition—Past and Present

Participation in non-professional endurance events continues to increase in popularity among adults in the United States. Regardless of the modality (e.g., cycling, running, or swimming), endurance training offers the participant a host of benefits for cardiometabolic health and cognition. Some of these benefits include a reduction in excess adipose tissue, reductions in systemic and chronic low-grade inflammation, improved metabolic flexibility, and a decreased risk of dementia development [1,2]. Interestingly, endurance training and sports nutrition appear to have synergistic and additive effects, as various nutritional strategies also offer protection to cardiometabolic health and can improve performance as well as maximize training adaptations [3]. Although earlier sports nutrition research largely focused on meeting an athlete’s training needs by ensuring sufficient carbohydrates were being consumed during the peri-workout window and an effective hydration plan was in place, today’s research has expanded quite a bit to include nutritional approaches and physiological outcomes that were ignored in these earlier years. Today, sports nutrition now also focuses on optimal recovery strategies, enhancing the immune system and reducing muscle damage following exercise, and periodizing nutrition for the individual rather than a “one size fits all” approach [4]. While coaches and athletes alike place significant attention on meeting the energy demands of their respective sport via manipulation of macronutrients and overall caloric load, a growing body of evidence suggests that athletes of all backgrounds have reported supplementing with some form of dietary supplement that includes micronutrients, polyphenols, and antioxidants, as well as various ergogenic aids [5].

### 1.2. Prevalence of Dietary Supplement Use among Athletes

Dietary supplement use is prevalent among athletes, with studies indicating that elite athletes tend to use dietary supplements more frequently than non-elite athletes [6]. Overall, the prevalence of supplement consumption among athletes ranges from approximately 40–60%, with proteins and multivitamins being the most commonly consumed supplements [5]. Despite the benefits sought by athletes, such as performance enhancement and recovery support, the prevalence of supplement use also raises concerns regarding the potential risks associated with adulterated or mislabeled products. To assist interested readers, the International Society of Sports Nutrition routinely publishes a thorough review aimed at maintaining an updated list of common dietary supplements used in athletics [7]. Still, no single article review can cover the wide range of dietary supplements that exist or provide an extensive discussion surrounding each supplement’s mechanisms of action. New dietary supplements are introduced to the sports nutrition market annually and therefore, reviews of individual dietary supplements can serve as an educational tool for sports nutritionists, coaches, and athletes for determining the safety and efficacy of individual supplements.

Among the various dietary supplements on the market today, exogenous antioxidants appear to be a common addition to an athlete’s ‘supplement stack’ and have garnered support for their potential protective effects against oxidative damage and inflammation following strenuous bouts of exercise [8]. One dietary supplement garnering attention specifically among endurance athletes is astaxanthin (AX), a powerful antioxidant that has demonstrated an ability to specifically target the mitochondria and serve there as a regulator of energy metabolism, potentially improving endurance performance [9,10]. However, an individual’s training and competition goals must be considered when deciding to supplement with antioxidants. While antioxidant supplements are used among athletes to protect against exercise-induced oxidative damage, several studies have documented that exogenous antioxidants may impair signaling to the stress-activated protein kinases and transcription factors that lead to gene transcription and muscle adaptation [11]. Currently, there are arguments for and against the use of dietary antioxidants among athletic populations, and consideration of the context is needed to make informed supplementation suggestions/decisions.

### 1.3. Salmon—Nature’s Ultimate Endurance Athlete

Endurance exercise is generally defined as anything lasting longer than 30 min. However, interest has grown among the endurance community in ultra-endurance events (any event lasting longer than 4 h), such as ultramarathons. These events aim to challenge the limits of human physiology and metabolism during prolonged exercise. Ultramarathons, for example, may include athletes covering ~161 km in a 30 h duration [12]. It is also common for the cardiovascular system to experience insult across such demanding physical tasks, including damage to the ventricular valves and elevations in cardiac troponin T [12,13], as well as chronic exposure to reactive oxygen species (ROS) and oxidative stress during the event [14]. While endurance exercise is often cited as a modality for improving metabolic health [15], ultra-endurance exercise may expose the athlete to excessive oxidative stress, resulting in an elevated risk of developing cardiovascular disease [14]. Although completing an ultra-endurance event is quite impressive, they pale in comparison to the distances covered by other migratory animals, such as salmon.

Salmon are revered as unparalleled endurance athletes due to their extraordinary physiological adaptations and remarkable feats of strength and stamina which can be observed during the migratory “salmon run”. Upon returning to their birthplace to spawn, salmon can travel ~1400 km upstream and against strong downstream rapids and currents. These migrations require sustained physical exertion over prolonged periods, highlighting the salmon’s remarkable aerobic capacity and muscular endurance. Moreover, salmon stop feeding during this migration, suggesting that the majority of their metabolic fuel is coming from endogenous lipid stores. Like humans, animals can also experience oxidative stress, and exposure to excessive ROS is known to impair mitochondrial function and lipid metabolism and hamper muscle endurance performance [16]. From an endurance perspective, salmon are an interesting species to study as they are able to complete the ultimate endurance event while still preserving an ability to utilize and oxidize lipids even in the presence of excessive exposure to ROS. This may be explained by the salmon’s high concentration of AX found in their muscle [17]. The accumulation of AX within phospholipid bilayers is well documented [18], and AX effectively mitigates lipid peroxidation during endurance activity [19]. The powerful antioxidant properties of AX may be a primary determinant in mitigating ROS formation during the salmon run and these observations have led to implementing AX supplementation as one intervention for possibly mitigating oxidative damage following endurance exercise [19]. While the exact mechanisms exerted by AX are not still fully understood, research is currently examining AX as a potential dietary supplement for exercising humans.

### 1.4. What Is Astaxanthin?

Astaxanthin is a red-orange and lipid-soluble keto-carotenoid that belongs to the terpenes class of chemical compounds. First discovered in 1938 [20], AX was originally used extensively as a natural ingredient in aquatic feed before researchers later discovered that AX possessed strong antioxidant properties due to its unique molecular structure [21]. At each end of its β-ionone rings, AX contains polar regions and a non-polar middle with 13 conjugated double polyunsaturated bonds (see Figure 1). Like other xanthophyll carotenoids (e.g., lutein, zeaxanthin), AX bears hydroxyl groups; however, its β-ionone rings have hydroxyl groups at the 3,3′-positions and keto groups at the 4,4′-positions. Moreover, its elongated structure allows it to permeate the membranes of a cell, allowing AX to neutralize free radicals and protect cells from oxidative stress within and outside of the cell [22]. Unlike other carotenoids which can exhibit pro-oxidative properties (lycopene and β-carotene), AX is deemed a “pure antioxidant” as it only exhibits antioxidant properties without the ability to convert to a pro-oxidant [23]. While AX can be found in certain fungi, as well as consumed from specific aquatic organisms such as krill, shrimp, and salmon, it is primarily supplemented through a microalgae (*Haematococcus pluvialis*), allowing a person to achieve the AX range of 4–12 mg/day often implemented in research to obtain potential health and performance benefits [24,25].

After its discovery and approval as a dietary supplement, a series of animal studies followed demonstrating the biological and antioxidative properties of AX [18]. While the mechanisms of action in AX were beginning to be better understood in the 1990s, it was not until the early 2000s that AX started being studied as a potential dietary supplement for enhancing exercise performance in animals [26,27] and humans [28]. However, it was almost a decade later (~2010) that AX demonstrated its potential for enhancing endurance performance when Earnest et al. (2011) showed a clear benefit in a 20 km cycling time trial in comparison to a placebo (PLA) [29]. In the present day, the last ~10 years of AX research have revealed several unique properties of AX that are not observed among other antioxidants. It is also clear now that a primary target following AX supplementation is the mitochondria [10]. This may then explain why null effects have been observed with AX in resistance training [28], but improvements have been found in events such as the 20 km time trial [29]. With regard to performance enhancement, it appears that individuals training for endurance events may benefit the most from AX supplementation. While a few well-written reviews on the benefits of AX supplementation and exercise performance exist [9,24], the purpose of this review is to briefly outline the training adaptations that follow from endurance exercise and then detail the role of AX in mitochondrial regulation, specifically as it relates to potentially improving endurance performance and enhancing mitochondrial training adaptations in the endurance athlete.

## 2. Exercise-Induced Hormesis

### 2.1. Finding the Right Balance

Exercise-induced hormesis refers to the phenomenon where low to moderate levels of stress from physical activity, such as endurance exercise, stimulate adaptive responses in the body, leading to improved physiological function and performance [30]. Endurance exercise, characterized by prolonged and repetitive movements, can induce the excessive production of ROS which can potentially overwhelm the mitochondria and cells, resulting in oxidative stress [31]. Oxidative stress is the result of an imbalance in the redox environment such that the production of ROS overwhelms antioxidant defense mechanisms, resulting in oxidative damage to endogenous biomolecules. It is well accepted that the mitochondria are common sources of ROS and increased strain on the electron transport complex during mitochondrial respiration is likely to lead to oxygen/electron leakage from complexes I and III, leading to ROS formation, specifically the superoxide and hydroxyl radicals, and potentially oxidative stress [31].

Interestingly, stress in general has widely been associated with negative aspects and performance outcomes [32], even though over 100 years ago, work from Yerkes and Dodson demonstrated a U-shaped relationship (i.e., a positive effect) between moderate amounts of physiological arousal (i.e., stress) on physical performance [33]. It is therefore important to clarify that acute exposure to ROS is potentially beneficial, as moderate ROS exposure serves as a trigger to upregulate antioxidant defense mechanisms and initiate signaling cascades for mitochondrial biogenesis [31]. Thus, the same U-shaped relationship between stress and performance can also be applied to redox balance and the hormetic stress response from endurance exercise. In other words, excessive exposure to ROS leads to oxidative stress and chronic low-grade inflammation, while acute/moderate exposure to ROS is beneficial for facilitating favorable mitochondrial adaptations known as mitohormesis [34].

#### Too Much of a Good Thing: Sickness Prevalence among Endurance Athletes

Although acute exposure to low/moderate amounts of ROS and oxidative stress is beneficial for stimulating favorable training adaptations such as mitochondrial biogenesis, excessive ROS exposure observed during intense training periods leading up to competition can overwhelm the mitochondria’s endogenous antioxidant capacity, resulting in impaired immune system function and decreased resistance to infection [35,36]. ROS have been implicated in many aspects of the immune response to pathogens mainly related to innate immunity. Indeed, they have been proposed to be the common determinant of inflammasome activation (e.g., lymphocyte activation and signaling of various immune-related enzymes such as B- and T-cell receptors), which is critical in the inflammatory process and thus necessary for an efficient immune response. However, repeated and chronic exposure to ROS may also mediate lymphocyte dysfunction and has been identified as a contributing component in various immune-related diseases, such as multiple sclerosis or inflammatory bowel disease [37]. Understanding this relationship between ROS and immune responses is of particular importance to endurance-trained athletes as 9 out of 10 runners report experiencing some type of running-related injury or illness leading into a half or full marathon [38]. These findings are supported by Peters and Bateman, who previously reported a depressed immune system in 150 runners who were identified as having an increased prevalence of upper respiratory tract infection after a ~50 km run [39]. Along with the aforementioned findings, numerous dietary supplement interventions have been examined as potential methods for mitigating oxidative stress in endurance athletes, including supplementing with various exogenous antioxidants such as vitamins C and E, CoQ 10, and curcumin. While supplementing with exogenous antioxidants has demonstrated a capacity to help combat ROS generation during exercise, it is important to note that ROS do potentially serve a beneficial role depending on the magnitude of accumulation [31]. Moderate amounts of ROS generation, as produced from exercise, can trigger favorable skeletal muscle adaptations and signaling molecules to increase endogenous antioxidant status and drive mitochondrial biogenesis [31,40], while excessive production can be toxic to cellular function. Therefore, an interesting problem is presented to any serious athlete interested in supplementing with exogenous antioxidants. By supplementing with exogenous antioxidants, they may potentially enhance recovery and training time by mitigating injury and damage at the cell level. However, supplement with too high of a dose and the athlete risks mitigating exercise-induced training adaptations [41]. Since one of the primary mechanisms through which many exogenous antioxidants work is by quenching ROS and preserving mitochondrial (and those from other cellular sources) endogenous antioxidant reservoirs [42], alternative interventions that also target the mitochondria may prove useful in mitigating ROS and oxidative stress without potentially hampering exercise-induced training adaptations.

## 3. Mitochondria Dynamics during Exercise

### 3.1. Mitochondria—Still the Powerhouse of the Cell

Few cellular organelles receive as much attention as the mitochondria when endurance performance is the topic of focus. Mitochondria play a pivotal role in the adaptation to endurance exercise due to their fundamental function in aerobic metabolism and energy production [43]. Endurance exercise training induces mitochondrial biogenesis, resulting in an increase in mitochondrial density and capacity within skeletal muscle [43]. This augmentation enhances oxidative phosphorylation and ATP synthesis, crucial for sustaining prolonged physical activity. Additionally, endurance exercise promotes mitochondrial quality control mechanisms, such as autophagy and mitochondrial fusion–fission dynamics, ensuring the removal of damaged mitochondria and the maintenance of a healthy mitochondrial network [44]. Thus, the ability of mitochondria to adapt and optimize energy production pathways is essential for improving endurance performance and also makes the mitochondria a potential target of interest for enhancing endurance performance. Although an in-depth review of the many pathways that promote mitochondrial health following exercise is beyond the scope of the current paper, a brief review is provided here for the reader.

During exercise and specifically endurance training of sufficient duration (>60-min) and intensity (>45% VO_2max_), a cascade of events takes place which appears to be initiated by a decrease in ATP and an increase in both calcium appearance and free fatty acid availability. Collectively, these metabolites signal for several downstream transcription pathways to increase their activity, including AMP-activated protein kinase (AMPK), peroxisome proliferator-activated receptor gamma coactivator-1 alpha (PGC-1α), the family of peroxisome proliferator-activated receptors (PPARs), nuclear respiratory factors (NRFs), and antioxidant response elements (AREs), to name a few [45]. During endurance training, AMPK serves as a pivotal regulator of cellular energy homeostasis, playing a crucial role in coordinating metabolic responses to varying energy levels [46]. AMPK acts as a cellular energy sensor, activated by declining ATP levels or increasing AMP levels, which indicate an energy deficit. Once activated, AMPK orchestrates a cascade of signaling events aimed at restoring cellular energy balance. It achieves this by promoting ATP-generating pathways, such as glycolysis and fatty acid oxidation, while simultaneously inhibiting energy-consuming processes such as protein synthesis and cell proliferation [46]. Additionally, AMPK regulates mitochondrial biogenesis and function by phosphorylating PGC-1α at multiple sites, ensuring efficient energy production and utilization for future endurance training events [46]. Repeated activation of PGC-1α results in a noticeable effect on mitochondrial quantity and function, which has implications for endurance athletes. Aside from the training-induced changes in lipolytic enzymes which improve the utilization and transportation capacity of fatty acids, the increase in mitochondrial count provides the cell with more sites for fatty acid oxidation, which can be observed in any laboratory setting by a lower respiratory exchange ratio during submaximal exercise following chronic endurance training. Likewise, endurance training induces specific adaptations in skeletal muscle, including alterations in fiber type composition and metabolic pathways.

Among these pathways, PPARs, particularly PPARδ, play a significant role in these training adaptations. Studies have shown that endurance training leads to fibe-type-specific increases in PPAR-α protein content, particularly in type I fibers, contributing to enhanced oxidative capacity [47]. PPARδ activation has been linked to increased running endurance and improved muscle reprogramming, characterized by a shift in fuel utilization towards fatty acid oxidation [48]. Additionally, PPARδ mediates oxidative adaptation following exercise training, highlighting its intricate involvement in endurance exercise-induced metabolic changes. These findings underscore the importance of PPARs, particularly PPARδ, in the regulation of metabolic adaptations in endurance training, thereby influencing muscle performance and endurance capacity.

Lastly, endurance training is known to induce oxidative stress, which promotes the activation of NRFs [49]. Nuclear respiratory factors serve as a class of regulatory proteins that interact with mitochondrial gene expression and various nuclear transcription factors, specifically those involved in mitochondrial biogenesis and function, enhancing the capacity for aerobic metabolism [49]. Similarly, endurance training stimulates the activation of AREs, likely through exposure to reactive oxygen species [50], which are DNA sequences responsible for regulating antioxidant enzyme expression [51]. This activation leads to increased synthesis of endogenous antioxidant enzymes, such as glutathione peroxidase, superoxide dismutase, and catalase, which help mitigate exercise-induced oxidative damage. Together, NRFs and AREs coordinate cellular responses to endurance training, optimizing mitochondrial function and antioxidant defenses to enhance exercise performance and promote overall physiological adaptation.

### 3.2. Mitochondrial Endogenous Antioxidants—Beyond Energy Production

While mitochondria are considered the “powerhouses” of the cell, their role in metabolic health and performance extends beyond simply energy production. Although often ignored when discussing these unique organelles, the mitochondria’s role in combatting ROS is through a complex endogenous enzymatic system composed of various antioxidants. In brief, endogenous antioxidants play a critical role in maintaining cellular health by scavenging ROS and protecting against oxidative stress-induced damage to cellular macromolecules, including lipids, proteins, and DNA [31]. Within the cell, endogenous antioxidants, such as superoxide dismutase (SOD), catalase (CAT), and glutathione peroxidase (GPX), form an intricate defense system against ROS-induced damage and mitigate oxidative stress, thereby preserving mitochondrial integrity and function, especially during endurance exercise [31]. Redox imbalance, characterized by excessive ROS production and decreased antioxidant capacity, leads to mitochondrial dysfunction and cellular damage. While a small amount of cell damage is to be expected with any type of physical activity, excessive ROS exposure and macromolecule damage, as observed post-endurance exercise, can result in sickness, impaired recovery, and a slower return to training. Thus, endogenous antioxidants help maintain redox balance, ensuring optimal mitochondrial function.

In line with the hormetic theory described earlier, while prolonged endurance training could be considered detrimental, regular endurance exercise induces significant adaptations in the endogenous antioxidant system, enhancing the body’s ability to counteract exercise-induced oxidative stress and maintain cellular homeostasis [52]. Regular endurance training promotes the upregulation of endogenous antioxidant enzymes within skeletal muscle fibers by repeatedly exposing the mitochondria to moderate levels of oxidative stress during prolonged physical activity. Over time, the mitochondria respond with an increased synthesis and activity of antioxidant enzymes, which enables more efficient scavenging of ROS generated during exercise, thus reducing oxidative damage to cellular components [52]. Furthermore, endurance training promotes mitochondrial biogenesis and enhances mitochondrial function, with a net effect of more mitochondria and a greater number of endogenous antioxidants for future insults to mitochondria integrity. Overall, the adaptations to the endogenous antioxidant system from endurance exercise contribute to improved oxidative stress resistance and enhanced exercise performance. Given the evidence on the endogenous antioxidant system, it is no surprise that research is turning its attention towards mitochondria-targeted exogenous antioxidants, which may hold promise as a therapeutic intervention for mitigating oxidative stress where excessive ROS exposure may be experienced, such as during prolonged endurance training. By delivering antioxidants directly to the mitochondria, these compounds may more effectively mitigate oxidative damage and preserve mitochondrial function. Therefore, exogenous antioxidants could potentially serve as a targeted approach and offer a novel strategy to preserving or even enhancing mitochondrial integrity and function.

## 4. Astaxanthin: Nature’s Most Powerful Carotenoid

### 4.1. Astaxanthin as a Regulator of Mitochondrial Function

Carotenoids belong to a class of bioactive molecules synthesized by plants and photosynthetic microorganisms. Although more than 800 carotenoids exist, each carotenoid exhibits distinct characteristics that are primarily attributed to their molecular structure. Carotenoids consist of an extensive series of conjugated double bonds in the central part of the molecule, which imparts their vibrant colors ranging from yellow to red in various fruits and vegetables. The exception is AX, which is only found in red-colored seafood. Overall, carotenoids serve crucial roles in plants, acting as antioxidants and defending against ROS during energy transfer within the mitochondria and other chemical reactions [53].

Among the many carotenoids discovered, AX has specifically garnered the attention of researchers and athletes for its potential to enhance aspects of endurance performance, which likely reflects AX’s role in regulating mitochondrial function. As outlined earlier, the mitochondria are the sole producers of ATP, as well as the final destination for oxygen transportation and utilization. When the cell experiences periods of increased energetic demands such as during endurance training, the mitochondria experience a dramatic influx of oxygen. Most of this oxygen is reduced to water in a series of chemical reactions along the electron transport chain. However, a significant amount of oxygen also escapes during the one-electron reduction of molecular oxygen, known as superoxide [31]. Superoxide serves as the parent molecule for all other ROS formation, although superoxide itself is not relatively reactive. Still, understanding that more damaging and reactive ROS are first formed from superoxide makes superoxide an interesting target when the aim is mitigating oxidative stress. Aside from AX’s ability to serve as a strong antioxidant among various ROS and reactive nitrogen species (RNS), AX possesses a strong affinity for the superoxide molecule, as well as peroxyl radical intermediates [18]. By quenching superoxide and returning it to yield ground-state oxygen, while also scavenging and deactivating peroxyl radical intermediates, AX provides robust protection where lipid-rich membranes exist (e.g., mitochondria, cells, etc.) (Figure 2). Moreover, the mitochondria are especially sensitive to Ca^2+^ disturbances, such that following excessive ROS exposure and endoplasmic reticulum disruption, mitochondrial membranes become increasingly permeable to Ca^2+^ which results in a loss of mitochondrial function and oxidative phosphorylation efficiency. Over time, the mitochondria may begin to release pro-apoptotic factors to induce mitosis [54]. However, several investigations have demonstrated that AX treatment can reduce or even prevent the Ca^2+^ influx by preserving the structures of various calcium channels such as the ryanodine receptor, maintaining the endoplasmic reticulum membrane, and increasing the affinity of calcium-binding proteins and muscle tetanic proteins [55,56,57,58].

When considering the hormetic theory, a common criticism of antioxidant supplementation is often that antioxidant supplementation might diminish stress exposure to the cell, resulting in impaired training adaptations [59]. This is an important point for the athlete considering supplementing with antioxidants and this criticism has been documented following acute and chronic antioxidant supplementation [60]. However, context remains key for the purpose of this review, where endurance training and endurance events remain the focus. In the context of intense endurance training blocks or following the completion of an endurance event (e.g., a marathon or triathlon), oxidative stress and an increased susceptibility to illness have both been documented [38,61]. Considering that the metabolic demands from training are merely one source of stress placed upon the mitochondria, the aim of AX supplementation for the endurance athlete is to mitigate the overproduction of ROS and the subsequent harmful events that follow. This was shown by Aoi et al., who demonstrated that when the mitochondria are overloaded from excessive stressors (i.e., heavy exercise and high-caloric meals), AX may protect the mitochondria from oxidative damage, as well as preventing detrimental modifications to muscle proteins and reducing systemic inflammation [19,27]. Although these series of studies were conducted in a rodent model, there are implications for the endurance athlete, where their time training can be maximized and their time recovering can be reduced.

### 4.2. Astaxanthin and Mitochondrial Dynamics—A Synergistic Pairing

Several transcription factors were outlined earlier (e.g., PGC1-α, PPAR, AMPK, etc.) regarding endurance training’s effect on mitochondrial dynamics. Likewise, AX has demonstrated comparable effects among similar transcriptional factors, suggesting a synergistic effect may exist between AX and endurance training on mitochondrial regulation. In fact, numerous studies have shown that AX treatment can correct for abnormal gene expression and protein modification in the mitochondria, while simultaneously mitigating oxidative damage and improving metabolic flexibility in the organism [19,26,27,62,63,64]. It is important then to understand the additional roles exerted by AX, which extend beyond simply serving as an exogenous antioxidant.

Following supplementation, AX may serve as a trigger for upregulating mitochondrial signaling proteins, such as NAD-dependent deacetylases (SIRT1, SIRT3). This family of sirtuins is known to increase the activity of various endogenous antioxidants, including heme oxygenase 1, superoxide dismutase, and catalases through its subsequent activation of forkhead boxes 01 and 3 (FOXO1, FOXO3) and nuclear factor erythroid 2-related factor 2 (Nrf2) [65]. Nrf2 is of particular interest for the endurance athlete as it is considered a master regulator of endogenous antioxidant responses by binding to AREs and through its increase in mitochondrial biogenesis by upregulating several mitochondrial-related genes [66]. Although other review articles exist that have outlined the effects of AX on Nrf2 [67], it is clear that AX supplementation increases the expression of Nrf2 in various rodent models [68,69]. Past work in rats demonstrated that AX increased tissue-specific levels of endogenous glutathione when exposed to oxidative damage induced by oral formaldehyde [70]. In agreement with these findings, McAllister and colleagues recently showed that AX supplementation significantly increased glutathione concentrations in healthy humans (~7%) without a subsequent decrease in markers of oxidative stress [71]. First, these are important findings regarding AX’s ability to regulate endogenous antioxidants as it was suggested that AX likely acted on the Nrf2 pathway to upregulate glutathione concentrations, possibly through the activation of γ-glutamyl cystine ligase [67]. Glutathione is often considered one of the most powerful endogenous antioxidant systems in the cell due to both its abundance and ability to revitalize other depleted antioxidants [72,73]. The second finding was the lack of changes in resting markers of oxidative stress (i.e., hydrogen peroxide and malondialdehyde) within a cohort of healthy, young men. AX supplementation has routinely demonstrated its ability to lower elevated markers of oxidative stress in populations at risk for cardiometabolic diseases such as insulin resistance and heart disease [74,75,76]. However, in a healthy population, lowering oxidative stress may not be beneficial as these various markers are also necessary in improving metabolism and cell responses when exposed at normal physiological levels. These findings were also observed by another recent study with AX supplementation, whereas AX demonstrated some protective properties of cardiometabolic health without affecting markers of muscle damage or inflammation [77]. Collectively, these findings suggest that AX may protect from an overproduction of ROS while maintaining physiological levels and correcting for potential abnormalities.

Similar to AX’s role in regulating Nrf2, AX supplementation appears to also mediate the activity of PGC1-α. While previous work has shown that AX supplementation will increase the expression of PGC1-α [78], it has remained unclear by which mechanisms AX controls PGC1-α expression. In a well-controlled study, Nishida et al. (2020) recently observed that in AMPK-knockout mice, PGC1-α activity remained absent following AX supplementation, suggesting that AX works through an AMPK-dependent pathway to mediate PGC1-α responses [63]. It is not surprising then that AX also likely exerts a beneficial effect on mitochondrial function by indirectly regulating mitochondrial fission/fusion through its mediation of AMPK and subsequent activation of dynamin-1-like protein (Drp1) [46]. Within skeletal muscle, Drp1 plays a critical role in ensuring adequate control of mitochondrial fission/fusion, such that in Drp1-knockout mice, endurance performance and training adaptations are significantly impaired [79]. Regardless of whether it is a direct or indirect effect, it is clear that AMPK plays a critical role in regulating energy balance and maintaining mitochondrial quality throughout all cells. Moreover, AMPK’s downstream effects, such as its ability to induce mitochondrial biogenesis through its regulation of PGC1-α gene expression, suggest that supplements such as AX may offer a metabolic and training advantage when dosed appropriately and supplemented chronically. When reviewing the evidence from AX supplementation in animal models with markers of endurance performance as the dependent variable of interest, the data are overwhelmingly positive [25]. While these findings from animal models do have important implications for the endurance athlete, similar studies in humans are currently limited and mixed, as detailed below.

## 5. Astaxanthin and Endurance Performance

### 5.1. A Review of Astaxanthin and Human Studies

During intense endurance exercise, several factors have been proposed in the etiology of fatigue, including oxidative stress, glycogen depletion, hydrogen ion accumulation, and increasing muscle acidity [80]. From a mechanistic standpoint and based on metabolic work in mice, AX supplementation has been shown to favorably affect each of these factors. As an antioxidant, AX may mitigate excessive oxidative stress exposure during exercise [19]. Metabolically, AX appears to also shift substrate metabolism towards fatty acid oxidation and away from carbohydrate oxidation, thus preserving endogenous glycogen stores as well as mitigating a rise in lactate and hydrogen ion accumulation, which should prevent a muscle-acidifying effect [27,81]. Based on these findings, a potential exists for AX to serve as an ergogenic aid specifically within the endurance-trained community (see Figure 3).

Among the human studies examined for the present review, to date, only three studies have tested AX supplementation’s effect on aerobic endurance performance, with all three studies using trained, male cyclists for their protocols [29,82,83]. Two revealed a performance improvement with AX supplementation [29,82], and one found no differences between the PLA or AX groups [83]. Due to the limited number of investigations, inferences and generalizations regarding AX supplementation on endurance performance remain mostly speculative. However, a closer look at these three studies reveals some interesting points for consideration in future studies. Starting with Brown et al. (2021), their team examined 12 mg/d of AX supplementation across a 7 d duration in a counterbalanced and crossover design [82]. In this study, participants completed a 40 km cycling time trial following each supplementation period, with completion time and substrate oxidation rates as primary markers of interest. Compared to PLA, AX supplementation significantly improved cycling time to completion (1.2%; ~50 s finishing difference), with greater fat oxidation rates in the AX group at completion of the 40 km [82]. Likewise, Earnest et al. (2011) supplemented well-trained cyclists with either AX (4 mg/d) or PLA for a 4-week period using a between-group design [29]. Following each group’s supplementation period, participants completed a 120 min pre-exhaustion cycling ride before participating in a 20 km cycling time trial, with time to completion in the 20 km and substrate metabolism as primary markers of interest. Among the AX participants only, a ~5% performance improvement was observed, as noted by a 2 min faster time to completion in the 20 km time trial, compared to the 18 s improvement in the PLA group. However, no differences were noted in the AX group from the pre- to post-supplementation period [29].

In contrast to the two studies which observed an ergogenic effect from AX supplementation, Res et al. (2013) observed no differences in cycling time trials or markers of substrate metabolism following 4 weeks of supplementation with either AX (20 mg/d) or PLA [83]. Interestingly, the two studies that did observe an ergogenic effect incorporated a lower dose of AX (4–12 mg/d; [29,82]) compared to the higher dose of 20 mg/d of AX in the latter study [83]. Strictly focusing on the potential ergogenic effects of AX, Brown and colleagues attributed their faster cycling time to an increased reliance on fat, rather than carbohydrates, as denoted by the higher fat oxidation rates towards the end of the 40 km time trial in the AX group [82]. Although mechanistic data were not collected in any of the three studies, theoretically, an increase in fat oxidation should have a glycogen-sparing effect and led to a reduction in fatigue-inducing metabolites, such as lactate. This speculation is supported by studies that have observed decreases in lactate following AX supplementation and various exercise protocols [84]. Similarly, Earnest and colleagues also observed a performance improvement from AX supplementation without a change in markers of substrate metabolism. These findings may suggest an alternative mechanism for eliciting an ergogenic effect from AX supplementation, although additional research is needed to confirm this.

Diverging from these findings, data remain which show that AX supplementation at much higher dosages (20 mg/d) in well-trained cyclists has no effect on the various markers of performance or metabolism [83]. Several explanations were provided for the lack of differences observed, including the high level of fitness displayed in the study (VO_2peak_ = 60 ± 1 mL·kg^−1^·min^−1^). The fitness level of participants is an important consideration for future studies. Since changes in mitochondrial control and quality remain a primary target with AX supplementation and subsequent changes in substrate metabolism, it is likely that in highly trained/elite endurance athletes, the mitochondria are so well developed that AX may have negligible effects at currently recommended doses. In other words, a “ceiling effect” may exist where mitochondrial changes, if any, induced by AX supplementation may be missed by laboratory equipment lacking the appropriate sensitivity to capture those changes (e.g., metabolic carts) in the highly trained athlete. Thus, AX supplementation may serve a more important role in the performance and metabolic changes of recreational endurance athletes, who make up the majority of the endurance population.

A second consideration relating to fitness level is the effects of aerobic fitness status on markers of substrate metabolism. Aside from exercise intensity, aerobic fitness is one of the strongest influencers of substrate metabolism during submaximal exercise [85]. This is an important consideration since parallel-group designs cannot account for this without group stratification. Therefore, this may explain why the only study to report substrate oxidation rate changes following AX supplementation was also the only study to incorporate a crossover design [82]. By having participants serve as their own control group, variances in substrate metabolism are accounted for and mitigate the variances that would be observed in a between-group design. Additional considerations such as testing standardizations and test-retest reliability have been detailed in a well-written review on AX supplementation and exercise metabolism and performance [24].

### 5.2. Alternative Mechanisms of Action—Looking beyond Performance

While more work is needed on AX supplementation and endurance performance, recent findings have shown some interesting effects that extend beyond cycling time trials (Table 1). For example, a recent study showed that 4 weeks of AX supplementation (12 mg/d) significantly improved subjective markers of recovery following an intense exercise-induced muscle-damaging protocol in resistance-trained males [86]. Although AX supplementation has not always demonstrated an ability to mitigate muscle damage or improve muscle soreness [28,77], this more recent study reported significant reductions in muscle soreness compared to PLA [86]. From a practical perspective, a reduction in non-invasive measures is meaningful for the coach and athlete, given that the subjective markers are suitable, are more affordable to measure in and outside of a laboratory setting, and do not impose further damage to tissues required for the collection of blood markers. Thus, AX may have implications for athletes involved in same-day, repeated bouts of competition (e.g., beach volleyball) or day-to-day competitions such as those seen in basketball tournaments.

Additional effects from AX supplementation include its ability to reduce exercising heart rate in active runners and overweight individuals [87,88], and possible effects on muscle-strength endurance [89]. The former finding is likely due to AX’s capability to modulate the sympathetic nervous system during exercise and to hyperpolarize cardiac cells [90]. For the endurance athlete, a lower submaximal heart rate during a given workload often denotes an ability to accomplish the same capacity of work with less effort. Reductions in heart rate during submaximal exercise are a common observation from chronic aerobic training studies and it appears that AX may offer some similar effect from a cardiovascular perspective. Regarding muscle-strength endurance, a group of college-age students who supplemented with AX (4 mg/d) for 6 months increased their repetitions to failure in the squat exercise by ~55%, compared to only a ~19% increase in the PLA group (*p* < 0.05; [89]). However, in the only other study to examine muscle-strength endurance with AX, no effect was found [86]. However, it should be noted that the second study on muscle-strength endurance incorporated resistance-trained males as participants [86], suggesting again that AX may exert a noticeable performance effect in the less-trained population.

Overall, it is clear that more research is needed in all of the aforementioned areas of performance, recovery, and cardiovascular and metabolic effects. While AX has primarily been examined for its role as an antioxidant, properties may exist that extend beyond simply mitigating oxidative stress and inflammation.

## 6. Conclusions

AX is a powerful antioxidant and is well-established within the in vitro literature. However, more work is needed in vivo, especially within the human model. To date, AX clearly demonstrates an ability to improve aspects of metabolic health and mitigate oxidative stress in the general population. However, when examining AX in the endurance athlete, the data are mixed and show either a performance improvement or a null effect. It is also important that the reader is aware that, unlike other dietary supplements which can demonstrate a performance decrement, this has not yet been reported as a result of AX supplementation. Collectively, the current findings suggest that AX may improve the following during exercise: substrate metabolism, endurance performance as measured by time to completion, lactate clearance, recovery, and exercising heart rate. However, more work is needed in each area before a clear conclusion can be drawn. It is also important to point out that none of the present AX studies in the current review were conducted in a female-only cohort. With the increasing attention on females in sport and the obvious physiological differences between males and females, AX supplementation in females is a completely unexplored area that hopefully will gain attention in the following years.

For endurance athletes interested in AX supplementation, the current recommendations for dosages range from 4–12 mg/d to elicit a potential ergogenic effect [24,25]. As outlined earlier, AX may exert a greater range of benefits for those recreationally trained compared to the elite endurance athlete, although this is simply a hypothesis drawn from the current literature. Present guidelines suggest a supplementation period of 3–4 weeks, based on animal models, although ergogenic and metabolic effects have been reported in as little as 7 days [82], with a 7 day washout period required to eliminate >99.9% of AX from circulation.

Finally, AX supplementation is relatively safe, with no negative side-effects reported in 87 human studies [91], and is not found on the 2024 National Collegiate Athletic Association for interested athletes and coaches. For future research, studies on females are needed across recovery, metabolism, and performance. For studies that continue to examine AX supplementation in males and in line with guidelines from Brown et al. (2018) [24], the performance protocols chosen should have high agreement in test-retest reliability. For studies examining changes in substrate metabolism, choosing homogenous cohorts, specifically as it relates to fitness training status, is important for better understanding AX’s role in modulating substrate metabolism during exercise. Lastly, when assessing recovery or metabolic health changes from AX supplementation, ensuring appropriate biomarkers are chosen based on the protocol implemented or primary metabolic outcome of interest (e.g., recovery, inflammation, oxidative stress, etc.) is important. Readers are directed to a comprehensive review on choosing appropriate biomarkers and protocols [92], which could be used with future AX supplement studies. Only once the aforementioned guidelines have been achieved will stronger conclusions be drawn about the potential for AX in the endurance community.

## Figures and Tables

**Figure 1 nutrients-16-01750-f001:**
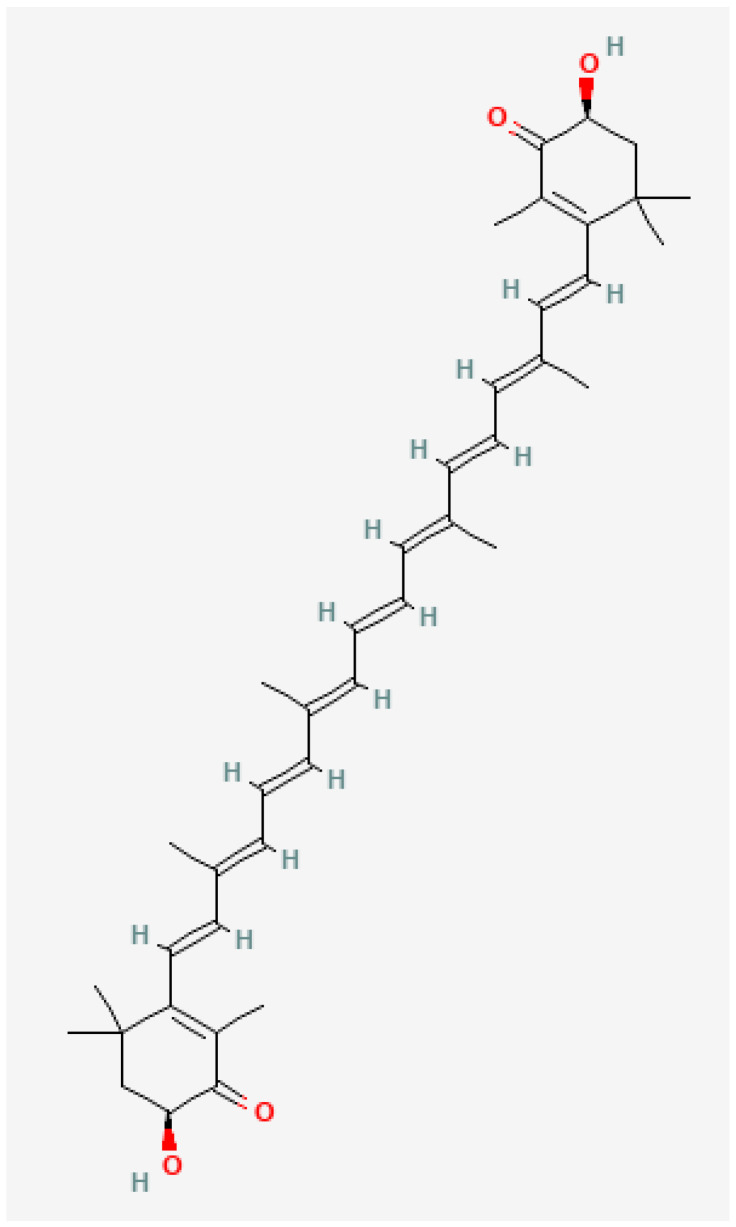
Astaxanthin’s structural formula.

**Figure 2 nutrients-16-01750-f002:**
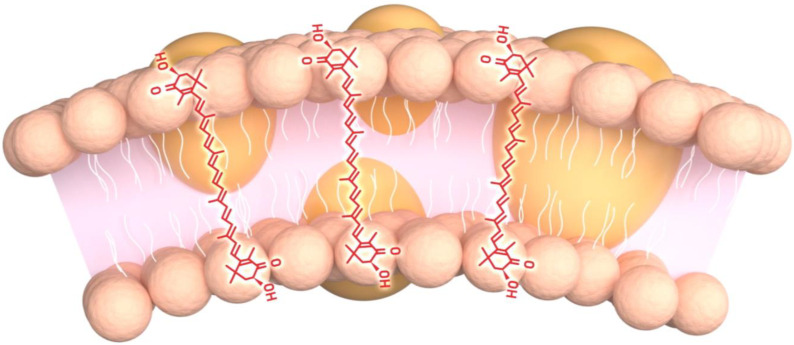
Astaxanthin possesses a hydrophobic conjugated polyene structure and terminal polar groups, effectively allowing it to quench free radicals within and outside of the mitochondrial membrane.

**Figure 3 nutrients-16-01750-f003:**
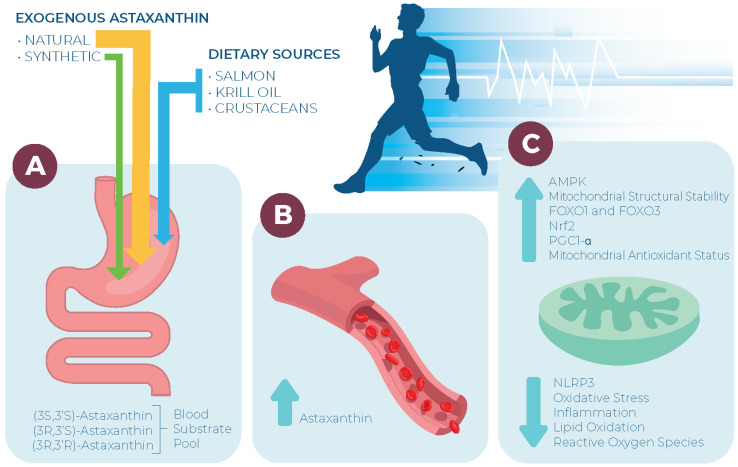
Pleiotropic effects of astaxanthin as a signaling molecule within the mitochondria. (**A**) Arrows from exogenous and endogenous sources denote suggested effect size on markers of interest for researchers; (**B**) Supplementing with endogenous and exogenous astaxanthin sources all markedly increase blood astaxanthin appearance; (**C**) Effects of astaxanthin on the mitochondria. The mitochondria are often only known as the ‘powerhouse’ of a cell due to the reliance on the mitochondria for adenosine triphosphate production. However, the mitochondria serve as an important focal point in improving cellular health and performance. An elevation in astaxanthin can contribute to the overall health of the mitochondria and astaxanthin has been shown to mitigate oxidative stress and upregulate antioxidant enzymes and anti-inflammatory transcription factors. These effects make astaxanthin a favorable metabolite for future projects with an aim to improve endurance performance in various populations. NLRP3, NOD-, LRR- and pyrin domain-containing protein-3; AMPK, AMP-activated protein kinase; FOXO1 and 3, forkhead box O1 and 3; Nrf2, nuclear factor-erythroid 2-related factor-2; PGC1-α, Peroxisome proliferator-activated receptor gamma coactivator 1-alpha.

**Table 1 nutrients-16-01750-t001:** Review of the Referenced Literature.

Author (Year)	Study Design	Participants	Intervention	Primary Outcome	Results
R.J. Bloomer [28]	Double-blind, Parallel design	Resistance-trained males	4 mg/d, 3 weeks	Muscle performance	No difference
C.P. Earnest [29]	Double-blind, Parallel design	Amateur endurance-trained males	4 mg/d, 4 weeks	Time trial performance	Improved performance
P.T. Res [83]	Double-blind, Parallel design	Well-trained endurance males	20 mg/d, 4 weeks	Time trial performance	No difference
Waldman [77]	Double-blind, Crossover design	Resistance-trained males	12 mg/d, 4 weeks	Muscle damage	No difference
Brown [82]	Double-blind, Crossover design	Endurance-trained males	12 mg/d, 7 d	Time trial performance	Improved performance
Barker [86]	Double-blind, Parallel design	Resistance-trained males	12 mg/d, 4 weeks	Subjective recovery	Improved recovery
Talbott [87]	Double-blind, Parallel design	Recreationally-trained males and females	12 mg/d, 8 weeks	Cardiorespiratory function	Improved cardiorespiratory function
Wika [88]	Double-blind, Parallel design	Overweight males and females	12 mg/d, 4 weeks	Substrate metabolism	Improved substrate metabolism
Malmsten [89]	Double-blind, Parallel design	Healthy college-age students	4 mg/d, 6 months	Muscle performance	Improved muscle performance

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
