# Peer review of "Astaxanthin Supplementation as a Potential Strategy for Enhancing Mitochondrial Adaptations in the Endurance Athlete: An Invited Review"

_nutrients, 2024, doi:10.3390/nu16111750_

Round 1

Reviewer 1 Report

Comments and Suggestions for Authors

This review pertains to relatively novel ergogenic aid from a specific nutrient, or carotenoid, astaxanthin.  The paper is organized and reads well. Given the novelty or recency of pertinent data, this is a review that the Nutrients audience will likely find interesting.

That said, there are few questions and suggestions that will likely improve this review:

1) a table of the studies to date pertaining to supplementation (dose they used, performance outcomes, participant group, any biomarkers, etc.)

2) line 209, what intensity and/or duration of exercise training is needed to elicit that response?  Low intensity for less than an hour may not be sufficient to achieve the "cascade of events" described.

3) There were a few places where a discussion was made regarding "excessive production of ROS" (lines 266 & 346).  Are there data that indicate the production is excessive or over-produced? Or, are the ROSs produced at a standard rate for that level exercise, it is an issue of insufficient resources (enzymes, compounds) to manage the oxidative state of the organelle or cell?  

4) Line 169 -- provide conext/content linking ROS production with immune function.  This paragraph seems to indicate that cytokine and immune function are the same as oxidation.  More clarity on the mechanism for oxidation affecting inflammation and immune function would be helpful.

5) Line 185 - there is discussion about energy balance and caloric intake, please provide data and reference for weight loss affecting ROS and leading performance impairment -- as the next sentence states that addressing ROS/oxidation may via "alternative interventions".  It seems there is an inference that the impaired performance with macronutrient changes and weight loss is due to ROS/oxidation. 

Comments on the Quality of English Language

The quality of English is sufficient to this reviewer. 

Author Response

Reviewer 1

Thank you reviewer, for your time and efforts in making this a better manuscript.

This review pertains to relatively novel ergogenic aid from a specific nutrient, or carotenoid, astaxanthin.  The paper is organized and reads well. Given the novelty or recency of pertinent data, this is a review that the Nutrients audience will likely find interesting.

That said, there are few questions and suggestions that will likely improve this review:

1) a table of the studies to date pertaining to supplementation (dose they used, performance outcomes, participant group, any biomarkers, etc.)

This has been created and added.

2) line 209, what intensity and/or duration of exercise training is needed to elicit that response?  Low intensity for less than an hour may not be sufficient to achieve the "cascade of events" described.

Valid point. A review of some of these down stream effects from exercise intensity/duration has been reviewed here (https://www.ncbi.nlm.nih.gov/pmc/articles/PMC2887994/) and added to the paper.

3) There were a few places where a discussion was made regarding "excessive production of ROS" (lines 266 & 346).  Are there data that indicate the production is excessive or over-produced? Or, are the ROSs produced at a standard rate for that level exercise, it is an issue of insufficient resources (enzymes, compounds) to manage the oxidative state of the organelle or cell?  

Ah, good perspective. The argument could be made either way, but is most often explained as excessive ROS since it’s the continual production of ROS that depletes these endogenous enyzmes/compounds resulting in OS. However, I see the point you are making as well and it is just as valid. But in keeping with the literature on this topic (see Powers ROS review from the present paper), describing it as excessive ROS is acceptable.

4) Line 169 -- provide conext/content linking ROS production with immune function.  This paragraph seems to indicate that cytokine and immune function are the same as oxidation.  More clarity on the mechanism for oxidation affecting inflammation and immune function would be helpful.

Additional content/context has been provided to give the reader additional clarity on the relationship between ROS and immune responses.

5) Line 185 - there is discussion about energy balance and caloric intake, please provide data and reference for weight loss affecting ROS and leading performance impairment -- as the next sentence states that addressing ROS/oxidation may via "alternative interventions".  It seems there is an inference that the impaired performance with macronutrient changes and weight loss is due to ROS/oxidation. 

Thank you for this comment. After re-reading this section, I simply removed it. It appeared to pull focus from the supplement being discussed to then turn attention towards dietary interventions. So sentences have been removed to keep the focus on astaxanthin’s effects.

Reviewer 2 Report

Comments and Suggestions for Authors

The paper submitted by Hunter Waldman provides a detailed summary of the role and application of astaxanthin in endurance training, and describes its various adaptations at the mitochondrial level.The review is interesting and well-down.

Author Response

Reviewer 2

The paper submitted by Hunter Waldman provides a detailed summary of the role and application of astaxanthin in endurance training, and describes its various adaptations at the mitochondrial level. The review is interesting and well-down.

Thank you for the kind words and time in reviewing the current manuscript. It was much appreciated.

Reviewer 3 Report

Comments and Suggestions for Authors

My feedback:

Abstract:

·         The goal of the review, which focuses on how AX supplementation affects mitochondrial adaptations in endurance athletes, is succinctly outlined in the abstract. To instantly notify the reader of the review's scope and attitude, it would be beneficial to begin with a better description of the key thesis or hypothesis.

·         The abstract gives some background information on AX and its origins, however it may improve the reader's comprehension if it quickly discussed the significance of these mitochondrial adaptations for endurance athletes. This would establish a stronger connection between the physiological impacts of AX and its usefulness in sports nutrition.

·         The description of the mitochondrial adaptations is somewhat vague. The abstract mentions "various adaptations which take place specifically at the mitochondrial level as a result of chronic endurance training," but does not specify these adaptations. A brief mention of key adaptations (like increased mitochondrial biogenesis or improved oxidative capacity) would be useful for setting the context.

·         The abstract makes the case that AX and endurance training may work in concert and provides helpful advice. It does not, however, offer any insight into the nature of these synergistic effects or what the data suggests. It would greatly increase the usefulness and clarity if the key conclusions or results from the review were summarized.

·         Include at least one significant finding or conclusion to strengthen the abstract; for example, "Recent studies suggest astaxanthin may enhance mitochondrial efficiency and reduce oxidative stress in endurance athletes, potentially improving performance and recovery."

1. Introduction

·         By discussing the past and present viewpoints in sports nutrition research, the Introduction skillfully demonstrates the significance of nutrition in endurance sports. The shift from general sports nutrition to the particular focus on AX, meanwhile, may be more seamless. To assist direct the reader's expectations and preview the special focus on AX, it can be beneficial to include a more explicit comment early in the section.

·         This section gives a thorough overview of the changes in sports nutrition research, highlighting the move away from macro-focused tactics and toward more sophisticated ones that take into account antioxidants and micronutrients. Although the historical background is acknowledged, the paper might make a stronger case for concentrating just on AX by outlining its distinct advantages over other antioxidants early in the Introduction.

·         The subsections are well-organized, moving from a general discussion of sports nutrition to more specific details about dietary supplements and the unique demands of endurance sports. However, the scope of the Introduction is quite broad.

·         The literature cited supports the statements, showing a robust connection between dietary strategies and athletic performance. However, the Introduction could benefit from including more recent studies or meta-analyses that highlight the specific effects of antioxidants on endurance training, which would bolster the premise of focusing on AX.

·         The technical content is accurate, with appropriate references to studies supporting the benefits of nutrition in endurance settings. However, the Introduction could discuss potential controversies or criticisms related to antioxidant supplementation in sports. This would balance the discussion and align it with critical scientific discourse, preparing the ground for addressing these issues later in the review.

2. Exercise-Induced Hormesis

·         The well-organized subsections could benefit from clearer transitions to guide the reader through the narrative. For instance, the leap from general hormesis to specific pathologies experienced by endurance athletes (Section 2.1.1) feels abrupt. Integrating these subsections with a brief introductory sentence or paragraph could improve flow.

·         While the manuscript addresses the role of ROS and their dual effects, the discussion could be expanded to include more detailed mechanisms by which exercise influences redox balance at the cellular level. Additionally, specific examples of how these mechanisms are modulated in various types of endurance training could enrich the content.

·         The section on dietary interventions (towards the end of section 2.1.1) feels somewhat tangential and underdeveloped. If the intention is to connect dietary strategies to hormetic effects, a deeper dive into how these diets specifically impact mitochondrial redox homeostasis would be beneficial.

·         The manuscript frequently mentions "mitochondrial biogenesis" as a positive adaptation to exercise-induced ROS. While this is correct, more recent studies suggest nuanced roles of different types of ROS in signaling pathways that could be highlighted (e.g., differences between hydrogen peroxide and superoxide in activating transcription factors like PGC-1α).

·         Several claims are made regarding the benefits of acute ROS exposure; however, the literature citations are sparse here. Including more up-to-date reviews or meta-analyses could strengthen these claims.

·         The term "hermetic" appears to be a typographical error for "hormetic." This should be corrected throughout the document to maintain professional integrity and accuracy.

·         A few metabolic pathways (AMPK, NRF, ARE) and their role in endurance adaptations are briefly discussed. For readers who aren't familiar with molecular biology, a more thorough description of these pathways—possibly with simplified diagrams—might improve understanding.

·         The section concludes with implications for endurance athletes but lacks practical recommendations based on the hormesis concept. A discussion on how athletes might apply this knowledge to training—perhaps through specific training regimens that optimize ROS production for beneficial adaptations—would be a valuable addition.

·         Consider discussing the limitations of current research and potential risks of misapplying the hormesis concept in practical settings, such as the risks associated with overtraining or excessive antioxidant supplementation, which might blunt beneficial adaptations.

3. Mitochondria Dynamics During Exercise

·         A more thorough examination of the molecular cues that exercise specifically sets off that result in mitochondrial adaptations would enhance the usefulness of the interesting but cursory overview of mitochondrial biogenesis. Although the publication lists a number of important molecules and pathways, including NRFs, PGC-1α, and AMPK, a more thorough examination of the interactions between these pathways could help readers who are not familiar with the subject understand it better.

·         The manuscript covers the basics of mitochondrial dynamics, including fusion and fission processes. However, the discussion could be expanded to include recent findings on how these dynamics are altered under different physiological and pathological conditions related to exercise.

·         Discuss the implications of altered mitochondrial dynamics in aging athletes or in the context of metabolic disorders, which are relevant to the endurance community.

·         The section on mitochondrial antioxidants effectively highlights their importance in protecting mitochondrial integrity during exercise. However, the manuscript could benefit from including recent advances in understanding how these antioxidants are upregulated in response to exercise and their long-term effects on health and performance.

·         While the manuscript integrates a range of studies, there appears to be an opportunity to more critically assess conflicting findings in the literature, particularly concerning the effects of supplements like AX on mitochondrial function.

·         Provide a critical appraisal of the literature with discussions on the limitations of current studies and potential reasons for inconsistent results, such as differences in exercise protocols, dosages of supplements, and participant characteristics.

4. Astaxanthin: Nature's Most Powerful Carotenoid

·         The section does an excellent job of tying AX's physiological effects and mitochondrial biochemistry together. Of particular note are AX's antioxidant properties and its ability to improve endurance performance by modifying mitochondrial activities. Though it might be more smoothly transitioned from basic information about carotenoids to specifics on AX's impacts. More direct links between the biochemical characteristics of AX and its functional results in endurance settings would improve impact and clarity.

·         The manuscript could benefit from a more critical analysis of the available research, particularly highlighting the limitations of current studies and the variability in findings. For instance, while AX's potential benefits are highlighted, a balanced view would also discuss studies with null or conflicting results in more detail.

·         Some claims appear broad or lack specific supporting evidence from human studies, especially concerning long-term health benefits and performance enhancement. A more cautious or nuanced interpretation of the results, possibly framing them within the context of ongoing research, would be more scientifically rigorous.

·         More direct evidence from human clinical trials, especially those that control for variables such as diet, training status, and genetic background, would strengthen the arguments.

5. Astaxanthin and Endurance Performance

·         The transitions between different subtopics (e.g., mechanisms of action, human studies) within the section on endurance performance are somewhat abrupt. The narrative could benefit from smoother transitions that help guide the reader through the logical progression of ideas.

·         There is a tendency to introduce new concepts or results without sufficient background, which could confuse readers not familiar with the specific area of research. For example, the discussion on mitochondrial dynamics and AX’s influence could be prefaced with a brief explanation of mitochondrial relevance in endurance contexts.

·         While the review mentions a few studies involving AX supplementation in trained cyclists, it lacks a critical analysis of the study designs, participant characteristics, and whether these factors could have influenced the outcomes observed.

·         More discussion on the dosage and duration of supplementation in relation to performance outcomes would strengthen the review. For instance, differentiating between effects seen at varying dosages or over short vs. long-term supplementation could provide practical insights for athletes and coaches.

·         The section would benefit from a more detailed mechanistic explanation of how AX influences the biochemical pathways related to endurance. The current explanation is somewhat superficial and could be expanded to include more on the interplay between oxidative stress, inflammation, and muscle function.

Comments on the Quality of English Language

The manuscript is understandable and conveys complex scientific information effectively. However, occasional grammatical errors and awkward phrases could be streamlined for clarity and flow. Improving sentence structure, fixing grammatical mistakes, and ensuring consistency in terminology would enhance the document's readability and professional presentation.

Author Response

Reviewer 3

Thank you reviewer for your time and efforts in the current manuscript. I’ve read through all comments you have provided and have made some changes based on these comments. Overall, I truly appreciate the amount of time and effort this reviewer put into the current review paper. However, a large portion of the comments that discuss readability and formatting/structure appeared suggestive and where I interpreted this, I did not make changes in these instances. Among three separate reviewers, two identified the writing/format/structure to be well done, with the current feedback from Reviewer 3 requiring rather extensive re-formatting/structure of several sections for readability purposes. This was completed in some areas and left alone in others. Example:

the comment regarding the leap from 2.1 to 2.1.1 as for transition. I ended section 2.1 with a balance view between acute vs. chronic ROS exposure and then reitereated this view to begin 2.1.1 as the transition into how endurance athletes may experience more chronic ROS as observed in the immune compromised literature following these type of events. After much consideration, I have chosen to leave the current readability as is in some, but not all areas (please see sections such as 2.1.1 where I attempted to clarify this section further). It is a tough task to balance between depth of various topics and not detracting from the current topic at hand, which was AX supplementation in endurance athletes, and much of the additional details were appropriate for a separate review. The preceding sections are intended to be brief and not extensively deep, as the review was not intended to be a review strictly on molecular pathways, ROS nuances, etc. I have however, made several of the suggested changes including to the abstract, word replacement (i.e., hermetic to hormetic), attempted to make stronger connections between variables where suggested, added additional references, and provided information on the specific pathways outlined by the reviewer which can be found in section 3.1 and limitations/take-aways in section 5.1.  

Abstract:

  • The goal of the review, which focuses on how AX supplementation affects mitochondrial adaptations in endurance athletes, is succinctly outlined in the abstract. To instantly notify the reader of the review's scope and attitude, it would be beneficial to begin with a better description of the key thesis or hypothesis.
  • The abstract gives some background information on AX and its origins, however it may improve the reader's comprehension if it quickly discussed the significance of these mitochondrial adaptations for endurance athletes. This would establish a stronger connection between the physiological impacts of AX and its usefulness in sports nutrition. The description of the mitochondrial adaptations is somewhat vague. The abstract mentions "various adaptations which take place specifically at the mitochondrial level as a result of chronic endurance training," but does not specify these adaptations. A brief mention of key adaptations (like increased mitochondrial biogenesis or improved oxidative capacity) would be useful for setting the context.
  • The abstract makes the case that AX and endurance training may work in concert and provides helpful advice. It does not, however, offer any insight into the nature of these synergistic effects or what the data suggests. It would greatly increase the usefulness and clarity if the key conclusions or results from the review were summarized.
  • Include at least one significant finding or conclusion to strengthen the abstract; for example, "Recent studies suggest astaxanthin may enhance mitochondrial efficiency and reduce oxidative stress in endurance athletes, potentially improving performance and recovery."
  1. Introduction

By discussing the past and present viewpoints in sports nutrition research, the Introduction skillfully demonstrates the significance of nutrition in endurance sports. The shift from general sports nutrition to the particular focus on AX, meanwhile, may be more seamless. To assist direct the reader's expectations and preview the special focus on AX, it can be beneficial to include a more explicit comment early in the section. This section gives a thorough overview of the changes in sports nutrition research, highlighting the move away from macro-focused tactics and toward more sophisticated ones that take into account antioxidants and micronutrients. Although the historical background is acknowledged, the paper might make a stronger case for concentrating just on AX by outlining its distinct advantages over other antioxidants early in the Introduction. The subsections are well-organized, moving from a general discussion of sports nutrition to more specific details about dietary supplements and the unique demands of endurance sports. However, the scope of the Introduction is quite broad. The literature cited supports the statements, showing a robust connection between dietary strategies and athletic performance. However, the Introduction could benefit from including more recent studies or meta-analyses that highlight the specific effects of antioxidants on endurance training, which would bolster the premise of focusing on AX. The technical content is accurate, with appropriate references to studies supporting the benefits of nutrition in endurance settings. However, the Introduction could discuss potential controversies or criticisms related to antioxidant supplementation in sports. This would balance the discussion and align it with critical scientific discourse, preparing the ground for addressing these issues later in the review.

  1. Exercise-Induced Hormesis

The well-organized subsections could benefit from clearer transitions to guide the reader through the narrative. For instance, the leap from general hormesis to specific pathologies experienced by endurance athletes (Section 2.1.1) feels abrupt. Integrating these subsections with a brief introductory sentence or paragraph could improve flow. While the manuscript addresses the role of ROS and their dual effects, the discussion could be expanded to include more detailed mechanisms by which exercise influences redox balance at the cellular level. Additionally, specific examples of how these mechanisms are modulated in various types of endurance training could enrich the content. The section on dietary interventions (towards the end of section 2.1.1) feels somewhat tangential and underdeveloped. If the intention is to connect dietary strategies to hormetic effects, a deeper dive into how these diets specifically impact mitochondrial redox homeostasis would be beneficial. The manuscript frequently mentions "mitochondrial biogenesis" as a positive adaptation to exercise-induced ROS. While this is correct, more recent studies suggest nuanced roles of different types of ROS in signaling pathways that could be highlighted (e.g., differences between hydrogen peroxide and superoxide in activating transcription factors like PGC-1α). Several claims are made regarding the benefits of acute ROS exposure; however, the literature citations are sparse here. Including more up-to-date reviews or meta-analyses could strengthen these claims. The term "hermetic" appears to be a typographical error for "hormetic." This should be corrected throughout the document to maintain professional integrity and accuracy. A few metabolic pathways (AMPK, NRF, ARE) and their role in endurance adaptations are briefly discussed. For readers who aren't familiar with molecular biology, a more thorough description of these pathways—possibly with simplified diagrams—might improve understanding. The section concludes with implications for endurance athletes but lacks practical recommendations based on the hormesis concept. A discussion on how athletes might apply this knowledge to training—perhaps through specific training regimens that optimize ROS production for beneficial adaptations—would be a valuable addition. Consider discussing the limitations of current research and potential risks of misapplying the hormesis concept in practical settings, such as the risks associated with overtraining or excessive antioxidant supplementation, which might blunt beneficial adaptations.

  1. Mitochondria Dynamics During Exercise

A more thorough examination of the molecular cues that exercise specifically sets off that result in mitochondrial adaptations would enhance the usefulness of the interesting but cursory overview of mitochondrial biogenesis. Although the publication lists a number of important molecules and pathways, including NRFs, PGC-1α, and AMPK, a more thorough examination of the interactions between these pathways could help readers who are not familiar with the subject understand it better. The manuscript covers the basics of mitochondrial dynamics, including fusion and fission processes. However, the discussion could be expanded to include recent findings on how these dynamics are altered under different physiological and pathological conditions related to exercise. Discuss the implications of altered mitochondrial dynamics in aging athletes or in the context of metabolic disorders, which are relevant to the endurance community. The section on mitochondrial antioxidants effectively highlights their importance in protecting mitochondrial integrity during exercise. However, the manuscript could benefit from including recent advances in understanding how these antioxidants are upregulated in response to exercise and their long-term effects on health and performance. While the manuscript integrates a range of studies, there appears to be an opportunity to more critically assess conflicting findings in the literature, particularly concerning the effects of supplements like AX on mitochondrial function. Provide a critical appraisal of the literature with discussions on the limitations of current studies and potential reasons for inconsistent results, such as differences in exercise protocols, dosages of supplements, and participant characteristics.

  1. Astaxanthin: Nature's Most Powerful Carotenoid
  • The section does an excellent job of tying AX's physiological effects and mitochondrial biochemistry together. Of particular note are AX's antioxidant properties and its ability to improve endurance performance by modifying mitochondrial activities. Though it might be more smoothly transitioned from basic information about carotenoids to specifics on AX's impacts. More direct links between the biochemical characteristics of AX and its functional results in endurance settings would improve impact and clarity.
  • The manuscript could benefit from a more critical analysis of the available research, particularly highlighting the limitations of current studies and the variability in findings. For instance, while AX's potential benefits are highlighted, a balanced view would also discuss studies with null or conflicting results in more detail.
  • Some claims appear broad or lack specific supporting evidence from human studies, especially concerning long-term health benefits and performance enhancement. A more cautious or nuanced interpretation of the results, possibly framing them within the context of ongoing research, would be more scientifically rigorous.
  • More direct evidence from human clinical trials, especially those that control for variables such as diet, training status, and genetic background, would strengthen the arguments.
  1. Astaxanthin and Endurance Performance
  • The transitions between different subtopics (e.g., mechanisms of action, human studies) within the section on endurance performance are somewhat abrupt. The narrative could benefit from smoother transitions that help guide the reader through the logical progression of ideas.
  • There is a tendency to introduce new concepts or results without sufficient background, which could confuse readers not familiar with the specific area of research. For example, the discussion on mitochondrial dynamics and AX’s influence could be prefaced with a brief explanation of mitochondrial relevance in endurance contexts.
  • While the review mentions a few studies involving AX supplementation in trained cyclists, it lacks a critical analysis of the study designs, participant characteristics, and whether these factors could have influenced the outcomes observed.
  • More discussion on the dosage and duration of supplementation in relation to performance outcomes would strengthen the review. For instance, differentiating between effects seen at varying dosages or over short vs. long-term supplementation could provide practical insights for athletes and coaches.
  • The section would benefit from a more detailed mechanistic explanation of how AX influences the biochemical pathways related to endurance. The current explanation is somewhat superficial and could be expanded to include more on the interplay between oxidative stress, inflammation, and muscle function.